

# Modelling studies of HOM and its contributions to growth of new particles: comparison of boreal forest in Finland and polluted environment in China

Ximeng Qi[1,2,4], Aijun Ding[1,2*], Pontus Roldin[3,], Zhengning Xu[1,2], Putian Zhou[4], Nina Sarnela[4], Wei Nie[1,2],
Xin Huang[1,2], Anton Rusanen[4], Mikael Ehn[4], Matti P. Rissanen[4], Tuukka Petäjä[4,1], Markku Kulmala[4] and
Michael Boy[4]

[1]Joint International Research Laboratory of Atmospheric and Earth System Sciences, School of
Atmospheric Sciences, Nanjing University, Nanjing, 210023, China
[2]Collaborative Innovation Center of Climate Change, Nanjing, 210023, China
[3]Division of Nuclear Physics, Lund University, P.O. Box 118, 22100 Lund, Sweden
[4]Institute for Atmospheric and Earth System Research / Physics, Faculty of Science, University of
Helsinki, P.O. Box 64, 00014 University of Helsinki, Helsinki, Finland

*Correspondence to*: Aijun Ding (dingaj@nju.edu.cn)

**Abstract.** Highly oxygenated multifunctional compounds (HOM) play a key role in the growth of new
particles during new particle formation (NPF) events, but their quantitative roles in different environments
of the globe haven't been well studied yet. In this study, the HOM concentrations and NPF events at a
remote boreal forest site in Finland (SMEAR II) and a sub-urban site in the polluted eastern China (SORPES)
were simulated with the MALTE-BOX model, and their roles in NPF at the two distinctly different
environments are discussed. We found that sulfuric acid and HOM organonitrate concentrations in the gas
phase are significantly higher but other HOM monomers and dimers from monoterpene oxidation are lower
at SORPES compared to SMEAR II. The model can simulate the NPF events at SMEAR II with a good
agreement but underestimate the growth of new particles at SORPES, indicating a dominant role of sulfuric
acid and HOM originated from oxidation of aromatics in the polluted environment. This study highlight the
needs of molecular-scale measurements in improving the understanding of NPF mechanisms in the polluted
areas like eastern China.



# 1 Introduction

New particle formation (NPF), including the production of the molecular clusters and the subsequent growth of these clusters (Kulmala et al., 2014), is a global phenomenon and has been observed under different environmental conditions (Kulmala and Kerminen, 2008; Kulmala et al., 2004; Zhang et al., 2012).
NPF can influence climate by contributing to up to 50% of cloud condensation nuclei (CCN) (Merikanto et al., 2009; Sihto et al., 2011) and can have strong effects on air quality (Shen et al., 2011; Yu et al., 2010; Guo et al., 2014).

Sulfuric acid has been commonly considered as one of the main gas precursors of NPF (Kulmala and Kerminen, 2008;Zhang et al., 2012). Recently, it was found that HOM can participate in the initial steps of
NPF by stabilizing the sulfuric acid (Schobesberger et al., 2013;Riccobono et al., 2014;Kulmala et al., 2013). Most of the HOM dimers and the most oxidized monomers can be extremely low volatility organic compounds (ELVOCs) (Kurtén et al., 2016) and most likely contribute to the initial growth of newly formed particles (Tröstl et al., 2016). Ehn et al. (2014) showed that monoterpene oxidation is a strong source of HOM and that these HOM can explain the majority of the observed particle growth from 2 nm up to 50 nm
in boreal forest. Recent studies (Jokinen et al., 2015;Tröstl et al., 2016) showed that HOM can enhance atmospheric new particle formation and growth in most continental regions and increase the CCN concentrations by applying a constant monoterpene HOM yield (achieved from measurement) in a global model. Based on the HOM formation theory described by Ehn et al. (2014), a detailed HOM formation mechanism was applied (Öström et al., 2017; Roldin et al., 2018).

Currently, the role of HOM in NPF has been mainly studied in specific environment conditions with intensive observations available, such as the SMEAR II station in Nordic boreal forest (Ehn et al., 2014; Yan et al., 2016). However, understanding the mechanisms of NPF is particularly important from the perspective of air quality. As one of the most economically invigorating and densely populated countries, measurements of NPF events have been started since last decade in China (Shen et al., 2011; Wu et al.,
2007; Wang et al., 2017; Kivekas et al., 2009). Interestingly, the NPF events were observed frequently in heavily polluted environments in China (Qi et al., 2015; Kulmala et al., 2017; Wang et al., 2017). However, no measurements of HOM in China are reported until now and the understanding of the roles of HOM in NPF are very limited. The SORPES station at Nanjing is one of the "flagship" station in the domain of Pan-Eurasia Experiment (PEEX) (Kulmala et al., 2015; Lappalainen et al., 2016), providing a completely
different environment in comparison to the remote boreal forest. In this study, the NPF events at SMEAR II and SORPES, including the formation rates, growth rates and environmental conditions, were compared firstly. Then, by using the new version of the MALTE-BOX model, the precursor vapor gases (i.e. sulfuric acid and HOM) and NPF at two sites were simulated to deeply investigate the differences of NPF. This



modeling study will increase our understanding about NPF at an urban site in China and examines whether the nucleation and HOM formation mechanisms, which are intensively investigated at SMEAR II in Finland, can be used in polluted environment in China. In addition, applying a process model like MALTE-BOX, to simulate HOM concentrations and their contribution to the growth of newly formed particles at the two

selected sites with different environmental conditions, can validate whether a single HOM formation and nucleation mechanism could be appropriate in global models.

## 2 Materials and Methods

### 2.1 Sites and observations descriptions

SMEAR II station (Station for Measuring Forest Ecosystem-Atmosphere Relations) is located in

Hyytiälä, Finland (Fig. 1). The station is a boreal forest site, which is surrounded by a Scots pine forest with high monoterpene emissions. The SORPES station (Station for Observing Regional Processes of the Earth System) is located in Nanjing, eastern China (Fig. 1) (Ding et al., 2013; Ding et al., 2016). The station is a sub-urban site and about 20 km northeast of downtown Nanjing. The aerosol number size distributions, trace gases and meteorological parameters were continuously observed at both stations. Volatile organic

compounds (VOCs) were observed by PTR-MS at SMEAR II continuously at different altitudes. The HOM monomers (molecules with even mass in 300-450 Th), dimers (molecules with even mass in 452-620 Th), organonitrate (represented by seven major molecules, i.e. $C_7H_9O_8NNO_3^-$, $C_{10}H_{15}O_{6-11}NNO_3^-$) and sulfuric acid concentrations were measured at SMEAR II by CI-API-TOF (Jokinen et al., 2012) during spring 2013. At SORPES, VOCs were observed by GC-MS from September to October in 2014 (Xu et al., 2017). More

details about the two stations and measurements are described by Hari et al. (2013) and Ding et al. (2016).

### 2.2 Model descriptions

In this study we applied the MALTE-BOX model (the model to predict new aerosol formation in the lower troposphere), a zero-dimensional model, which includes several modules for the simulations of chemical and aerosol dynamical processes (Boy et al., 2006). This model has been successfully utilized in NPF

analysis - for instance, reproducing OH radical and gaseous sulfuric acid levels (Petäjä et al., 2009), validating various plausible nucleation mechanism and particle growth (Boy et al., 2007; Wang et al., 2013b), and identifying important factors influencing NPF occurrence (Boy et al., 2006; Boy et al., 2008; Ortega et al., 2012).

The gas-phase chemistry was simulated using the Master Chemical Mechanism version 3.3.1

(MCMv3.3.1, http://mcm.leeds.ac.uk/MCM/) solved by Kinetic PreProcessor (KPP) (Damian et al., 2002).





In addition, a new HOM autoxidation mechanism, which is constructed based on the oxidation of monoterpenes (Ehn et al., 2014), was added into the MCMv3.3.1. This HOM mechanism explicitly describes the HOM formation processes, i.e. ozone oxidation of monoterpenes, intramolecular H-shift and $O_2$ additions (autoxidation) (Öström et al., 2017; Roldin et al., 2018). Moreover, based on Molteni et al.

(2016), a simplified mechanisms of HOM formation from the oxidation of aromatics by OH were added into MCM3.3.1. The aerosol dynamical processes were simulated with the size-segregated aerosol model, UHMA (University of Helsinki Multicomponent Aerosol model) (Korhonen et al., 2004). A fixed sectional approach with 120 bins from 1 nm to 2.5 μm in diameter was used. For the smallest size bin, the formation rates of newly formed particles were estimated by the function of sulfuric acid and a first-generation

oxidation product of the included monoterpenes, i.e.

$$J_1 = k*[H_2SO_4][HOM_{nuc}],$$

where $HOM_{nuc}$ was formed with a molar yield of $10^{-5}$ for each monoterpene reacted with OH (Roldin et al., 2015). The kinetic coefficient ($k$-value) was set for each case to achieve the highest correlation compared to the measured newly formed particles. Organic compounds with pure liquid saturation vapor pressure less

than 0.01 Pa were chosen as condensing vapors in UHMA. The saturation vapor pressures of organic compounds in MCMv3.3.1 were estimated with the group contribution method by Nannoolal et al. (2008) using the UManSysProp online system (Topping et al., 2016). The saturation vapor pressures of HOMs were calculated by SIMPOL (Pankow and Asher, 2008) as Nanoolal et al. (2008) method produces unrealistic estimates of vapor pressures for multifunctional HOMs containing hydroperoxide or peroxy acid

group (Kurtén et al., 2016). $H_2SO_4$ was treated as a non-volatile condensing vapor, which generally is a reasonable assumption at typical atmospheric relative humidity and $NH_3$ levels (Tsagkogeorgas et al., 2017). The coagulation process, dry deposition process and the dilution of aerosol number concentration caused by boundary layer evolution were estimated in the model as well.

The measurement variables, i.e. meteorological conditions (e.g. air temperature, relative humidity,

pressure and radiation), trace gases concentrations (e.g. $SO_2$, $O_3$, NO, $NO_2$, CO) and VOCs (e.g. ethylene, ethane, propane, acetone, methyl vinyl ketone, n-Butane, benzene, toluene, o-/m-xylene, 1,2,3/1,2,4-trimethylbenzene, ethylbenzene, isoprene and monoterpenes), were input into the MALTE-BOX model every 10 min. As monoterpenes were not measured by GC-MS at SORPES, monoterpene concentrations at SORPES were simulated using WRF-Chem, following the method of Huang et al. (2016), in which it was

shown that the MALTE-BOX model worked well in NPF simulation with WRF-Chem output of VOCs. The measured aerosol number size distribution was read into the model during the first five hours. The



chemistry scheme was run with a spin up time of 24 hours, in order to achieve a realistic gas-phase composition before the aerosol module was switched on.

## 3 Results

### 3.1 Comparisons of NPF at SMEAR II and SORPES

According to long-term observations, the frequency of NPF at SMEAR II is 23%, with highest value in spring months (about 40-50%) (Nieminen et al., 2014). Although the concentration of pre-exiting particles is high, which inhibit NPF, the NPF occurs even more frequent in Chinese megacities such as Nanjing. The frequency of NPF at SORPES is 44%, with highest value also in spring month (55%) (Qi et al., 2015). As shown in Table 1, the averaged formation rate of 6 nm particles ($J_6$) at SMEAR II are 0.3 $cm^{-3}s^{-1}$ while the $J_6$ at SORPES is 2.3 $cm^{-3}s^{-1}$ in average, which is almost 8 times higher than at SMEAR II. The growth rate of 6-30 nm particles (GR) is also higher at SORPES, with 4.5 nm/h at SMEAR II compared to 8.7 nm/h at SORPES in average.

The environmental conditions during NPF at the two sites are substantially different. First, the pre-existing particle loading is much higher at SORPES than at SMEAR II. The CS at SORPES is almost 20 times higher than at SMEAR II (Table 1). High CS tends to inhibit the occurrence of NPF because of the scavenging of cluster and the loss of gas-phase low-volatility compounds (Kulmala et al., 2017). Secondly, the concentrations of atmospheric oxidant such as ozone are higher at SORPES (Table 1). Moreover, the concentrations of OH and $NO_3$ radical in YRD urban area of China are higher than the clean area (Wang et al., 2013a;Nan et al., 2017).Third, the anthropogenic pollutants are much higher at SORPES than at SMEAR II. As an important gas precursor of NPF, the $SO_2$ concentration at SORPES is almost 50 times higher than at SMEAR II (Table 1). The concentration of $NO_x$, which is believed to suppress the NPF by reacting with peroxy radicals (Wildt et al., 2014), is also much higher at SORPES.

To in depth study the differences of NPF at SMEAR II and SORPES, the four NPF days and one non-NPF day at each site were chosen for simulations with MALTE-BOX (Table 2). Besides the differences of NPF parameters and environmental conditions at the two sites described above, monoterpene and benzene concentrations on each day at the two sites are tabulated in Table 2. Because of the high monoterpene emissions in southern China (Fig 1), the monoterpene concentrations are relatively high at SORPES especially when the air masses origin from south. The averaged monoterpene concentration on chosen days is 0.05 ppbv at SORPES compared to 0.13 ppbv at SMEAR II. As a sub-urban site, the anthropogenic VOCs (e.g. benzene, Table 2) are higher at SORPES than at SMEAR II, with 0.54 ppbv of benzene concentration at SORPES compared to 0.06 ppbv at SMEAR II in average. The averaged concentration of aromatics



(including benzene, toluene, o-/m-xylene, 1,2,3/1,2,4-trimethylbenzene, ethylbenzene) at SORPES on chosen days was 1.2 ppbv.

**3.2 The differences of simulated condensing vapors at two sites**

As shown in Fig 2a, similar to previous studies (Zhou et al., 2014), the model underestimates the concentrations of sulfuric acid at SMEAR II especially at night. The reasons for this discrepancy could be that there are other oxidants besides OH and Criegee Intermediate radicals lead to the formation of sulfuric acid (Boy et al., 2013). Because of the detection limit of the CI-API-TOF, the HOMs non-nitrate monomers, dimers and organonitrates presented in Figs. 2b-d contain 7-14, 8-17, 7-14 oxygen atoms, respectively. The model predicts the measured diurnal cycle of HOM non-nitrate monomers at SMEAR II with good agreement. For HOM dimers, the simulated concentrations are higher than the measurements at night while slightly lower at daytime when the NPF events occur. For HOM organonitrate, although matching well with measurements at daytime, the simulation results have stronger diurnal pattern, with much lower concentrations than measurements at night. In general, the normalized mean bias (NMB) values of sulfuric acid, HOM non-nitrate monomers, dimers, organonitrates and total HOM are -63.0%, 11.1%, 174.3%, 8.0% and 38.3%, respectively. Considering the uncertainties of the CI-API-TOF in measuring gas HOM (estimated uncertainty up to a factor of 2-3) and the many unknowns in their formations, the model provides an acceptable agreement between simulated and measured vapor concentrations.

Although no measurements of sulfuric acid and HOM are conducted at SORPES, a comparison of the simulated gas vapors concentrations at two sites can help us to qualitatively understand the differences between the boreal forest and polluted areas in China. As shown in Fig. 2a, the simulated sulfuric acid at SORPES is one order of magnitude higher than at SMEAR II at daytime. The high value of sulfuric acid is mainly related to the extremely high $SO_2$ concentrations and high atmospheric oxidation capacity at SORPES. Such high simulated sulfuric acid concentration is consistent with the measurements conducted in other urban sites in China, e.g. about $10^7$ #/cm$^3$ in Beijing (Wang et al., 2013b). The simulated HOM non-nitrate monomer concentrations from monoterpene oxidation are lower at SORPES (Fig. 2b) because of low values of monoterpene concentrations and high condensation sink. The simulated HOM dimer concentrations are much lower at SORPES than at SMEAR II while HOM organonitrate concentrations at SORPES are one order of magnitude higher than at SMEAR II (Fig. 2c, d). It is mainly because high NO concentrations at SORPES suppress the HOM dimer formation but contribute to the formation of HOM organonitrates.

As the comprehensive HOM formation mechanisms used in MALTE-BOX model are only based on the oxidation of monoterpenes, the simulated HOM monomer, dimer and organonitrate concentrations





presented above only refer to the HOM formed from monoterpene autoxidation. However, recent lab experiment shows that the aromatic hydrocarbons (e.g. benzene, toluene, o-/m-/p-xylene, 1,3,5-/1,2,3-/1,2,4-trimethylbenzene) oxidized by OH can lead to a subsequent autoxidation chain reaction forming HOM, which is believed to contribute substantially to NPF in urban area (Molteni et al., 2016). Therefore,

according to Molteni et al. ( 2016), a HOM molar yield of 3% for the OH oxidation of the aromatic species was assumed and added into the MCMv3.3.1. The contributions of aromatics oxidation to the HOM can be ignored in the remote boreal forest because of extremely low aromatics concentrations. However, as shown in Fig 3, the HOM from aromatics oxidation at SORPES can be above $10^8$ #/cm$^3$, which is about one magnitude higher than HOM from monoterpene oxidation. HOM concentration from aromatics oxidation

on NPF days is obviously higher than non-event days, reflecting an important role of HOM in NPF. Such high concentration of HOM from aromatics oxidation is caused by the high levels of aromatics and OH radical in the polluted urban environment and may contribute substantially to the SOA formation.

### 3.3 The simulations of aerosol size distributions at two sites

Figure 4 shows the variations of measured and simulated aerosol number size distribution at SMEAR II

and SORPES. The kinetic coefficients (*k*-value) on each day at both sites (tuned to cover the observed particle formation rates) is texted in Fig 4b and 4d. For the SMEAR II site, the model can capture both the NPF events and non-NPF events with same *k*-value, i.e. $1 \times 10^{-18}$ m$^3$ s$^{-1}$. Comparing the observed and simulated formation rates of 6 nm particles at SMEAR II (Table 3), the model underestimated the formation rate on 1 May, 2013 but overestimated the formation on other NPF days. During event days, more than one

banana shape was simulated at SMEAR II, which is mainly because of the multi-peaks of simulated sulfuric acid. For SORPES station, the *k*-value is higher than at SMEAR II in average and with more discrepancies. The *k*-value on 22 September, 2014 is more similar with the value at SMEAR II but much lower than other chosen days. The variations of the *k*-values can reflect the variability of other unaccounted compounds involved in the particle or cluster formation and initial growth (Kuang et al., 2008). As only HOM from

monoterpene oxidation was involved in the nucleation mechanism of MALTE-BOX, the much higher *k*-values at SORPES except on 22 September, 2014 reflects that other compounds, which are most probable the oxidation products of anthropogenic pollutants, can also involve in the nucleation. Moreover, the model cannot simulate the high formation rates observed at SORPES except on 22 September, 2014 (Table 3).

For simulations at SORPES station, the brief formation mechanisms of HOM from aromatics were

30 added in the MCM and the saturation vapor pressure of HOM were calculated by SIMPOL. However, even if we decrease the pure liquid saturation vapor pressures of HOM from aromatics oxidation with 2 orders of magnitude, the model underestimates the growth during the event days, except on 22 September, 2014.





The simulated growth rates on 22 and 24 September, 4 and 6 October are 7.8, 3.3, 2.8 and 2.8 nm/h, compared to the observed growth rates with 9.9, 16.2, 14.9 and 12.9 nm/h, respectively (Table 3). These results indicate that under polluted environmental condition there must be some other important gas vapors that are not accounted for in the model that contributes to the growth. Tao et al. (2016) found that

heterogeneous uptake of amines can effectively contribute to particle growth of newly formed particles in polluted YRD area of China. Comparing the averaged observed and simulated number size distribution (Fig 5), the simulated aerosol size distributions were in good agreement with measurements at SMEAR II, but the simulated number concentrations in the size range below 200 nm at SORPES are extremely lower than the observation. One reason is that primary particle emission is an important source of ultrafine particles in

urban areas (Qi et al., 2015), but not accounted in the model. Another reason is that current chemistry mechanisms and the accounted VOCs in the model dramatically underestimate secondary organic aerosol (SOA) formation in polluted area.

Only the NPF event on 22 September, 2014 was simulated in good agreement with measurement because this day had the lowest condensation sink and highest aromatics concentrations among the chosen

NPF cases at SORPES. Fig. 6 presents the footprints of all the cases at SORPES. The air mass on 22 September, 2014 was from marine area. Previous study shows that these marine air masses have the lowest accumulation mode particles concentrations and therefore the NPF occurs frequently (Qi et al., 2015). Although having the lowest condensation sink, the aromatics concentration on this day was still quite high, which was most probably emitted from local petrochemical industrial area. The air masses on 24 September

and 6 October were from North China and brought air pollutants to Nanjing (Figs. 6b, 6e). On 4 October, it had similar retroplumes with those on 22 September but with more local origin (Fig. 6). Holiday effects in China (National Holiday with more family vacations during 1-7 October) caused the high $NO_x$ and anthropogenic VOCs concentrations on this day (Xu et al., 2017). The formation and growth of NPF were suppressed by high $NO_x$ concentration and therefore cannot be simulated by current MALTE-BOX model.

**3.4 The differences of relative contributions of precursor vapors to growth at two sites**

Figure 7 shows the averaged relative contributions of precursor vapors to the growth of sub-100 nm particles from 9:00 to 15:00 LT during the four chosen NPF days at SMEAR II and on 22 September, 2014 at SORPES. Only the NPF event on 22 September, 2014 was presented at SORPES because current MALTE-BOX model can only capture the shape of NPF on this day. At SMEAR II, the growth of ultrafine particles

was dominated by HOM from monoterpene oxidation, which is consistent with the previous study by Ehn et al. (2014). HOM monomers contribute most to the growth at SMEAR II as they have high concentrations and relatively low saturation vapor pressures.



The relative contributions of precursor vapors to the growth of particles at SORPES are quite different with those at SMEAR II. First, through the higher gas-phase sulfuric acid concentration at SORPES (as shown in Fig 2), sulfuric acid has huge contributions to the growth of ultrafine particles at SORPES while playing a minor role in the growth at SMEAR II. Second, high NO concentration at SORPES switches the

5 formation of HOM non-nitrate monomers and dimers to the formation of HOM organonitrates. As under the same oxygen to carbon ratio the saturation vapor pressures of organonitrates were higher than non-nitrate monomers and dimers, the HOMs from monoterpene oxidation contribute less to the growth at SORPES in general. Third, at SORPES, HOM from aromatics oxidation play a dominant role in the growth of ultrafine particles because of high aromatics concentrations. Dai et al. (2017) conducted the simultaneous

measurements near a petrochemical industrial area in Nanjing and found that the anthropogenic VOCs have significant contributions to both the nucleation and the growth. This is also consistent with the previous study at SORPES that higher growth rates were observed when the air masses were from the YRD area with high anthropogenic VOCs emissions (Qi et al., 2015).

**4 Conclusions**

Higher frequency, formation rates and growth rates of new particle formation (NPF) events were observed at SORPES, a sub-urban site in eastern China, compared to SMEAR II, a boreal forest site in Finland. To quantitatively understand the differences of NPF at the two sites, the condensing vapors (i.e. sulfuric acid and HOM) and particle number size distributions were simulated by a new version of MALTE-BOX model with the comprehensive HOM formation mechanism based on monoterpene oxidation and simplified

mechanism of HOM formation from aromatics oxidation.

The model was proved to work well on simulating the sulfuric acid and HOM from monoterpene oxidation by comparing them with measurements at SMEAR II. Comparing the simulated sulfuric acid and HOM from monoterpene oxidation at two sites, the sulfuric acid and HOM organonitrate concentrations were much higher while the concentrations of HOM non-nitrate monomers and dimers are lower at

25 SORPES than at SMEAR II. High concentration of HOM from aromatics oxidation were simulated at SORPES. The differences of gas vapors (sulfuric and HOM) at two sites are mainly because the substantially higher $SO_2$, NO, aromatics concentration and condensation sink at SORPES. The model can simulate the particle number size distributions on NPF and non-NPF days with same kinetic coefficient at SMEAR II. However, the $k$-value is more divergent at SORPES, which means the mechanism of nucleation

at polluted urban is more complicated. HOM from monoterpene oxidation contribute more to the growth at SMEAR II while the sulfuric acid and HOM from aromatics play dominant roles in the growth of newly formed particles at SORPES. This study highlights that sulfuric acid and HOM concentration and their





relative contributions to the growth are distinct at different environmental conditions. To better understand the formation mechanisms of secondary aerosols, sulfuric acid and HOM measurements in urban regions in China are needed.

*Data availability.* The data of SMEAR II station (including meteorological, trace gas, VOCs, aerosol size distribution) are available at https://avaa.tdata.fi/web/smart, and data of SORPES (meteorological, trace gas, VOCs, aerosol size distribution) are available upon request from the corresponding author before the SORPES database are opened publicly. Emission data are available at http://eccad.sedoo.fr/eccad_extract_interface/.

*Acknowledgments.* This study was supported by Ministry of Science and Technology of China (2016YFC0200500; 2016YFC0202000), the National Natural Science Foundation of China (41725020, 41505109, 41675145, 91544231), the European Research Council (ERC) under the European Union's Horizon 2020 research and innovation program (grant agreement number 638703(COALA), 742206 (ATM-GTP) and 227463 (ATMNUCLE)), and the Academy of Finland Center of Excellence program (272041). The authors would like to thank the CSC-China Scholarship Council for the joint Ph.D. grant and thank Dr. Theo C. Kurten for suggestions on the paper.

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




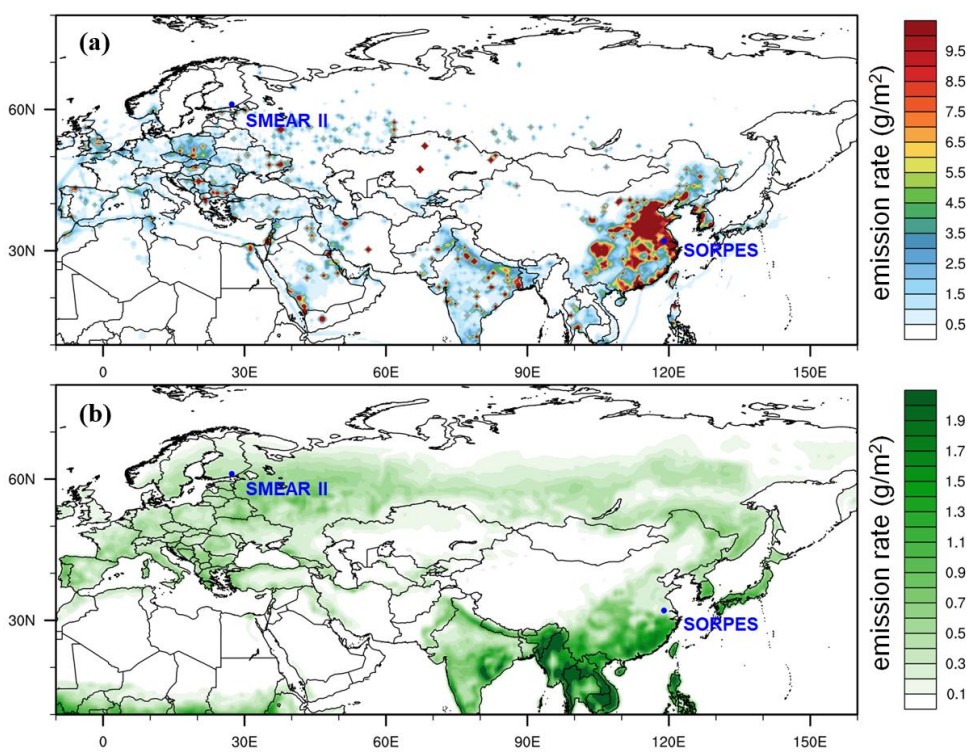

**Figure 1.** Sites (SMEAR II and SORPES) locations on map of emission inventory of (a) $SO_2$ and (b) monoterpenes.



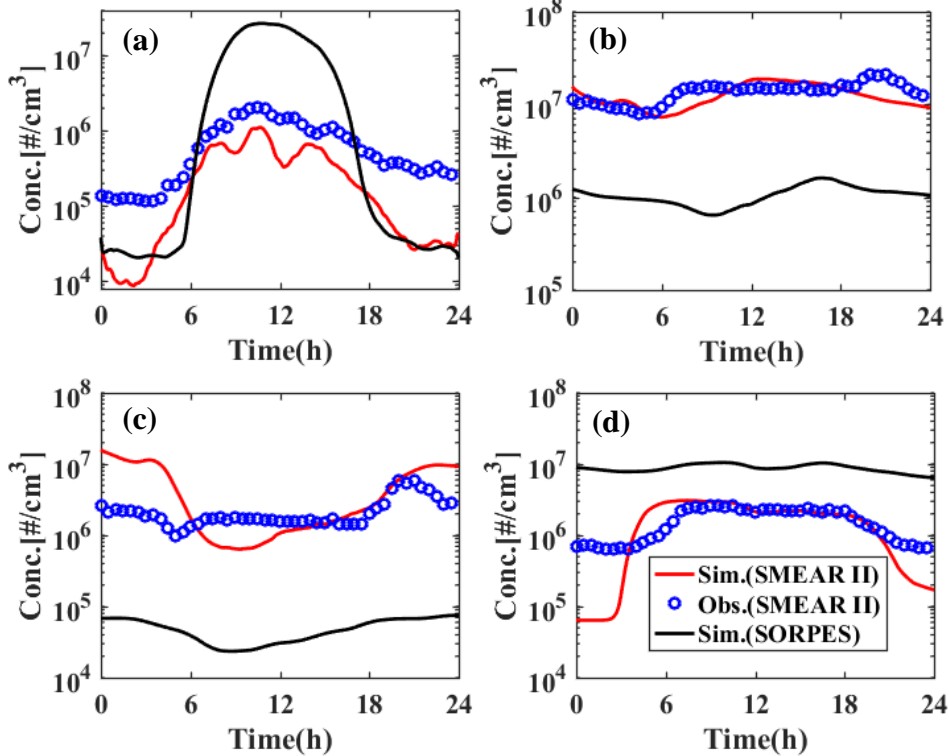

**Figure 2.** Averaged simulated and measured diurnal cycles of (a) $H_2SO_4$, (b) HOM non-nitrate monomers, (c) HOM dimers and (d) HOM organonitrates at SMEAR II and SORPES.





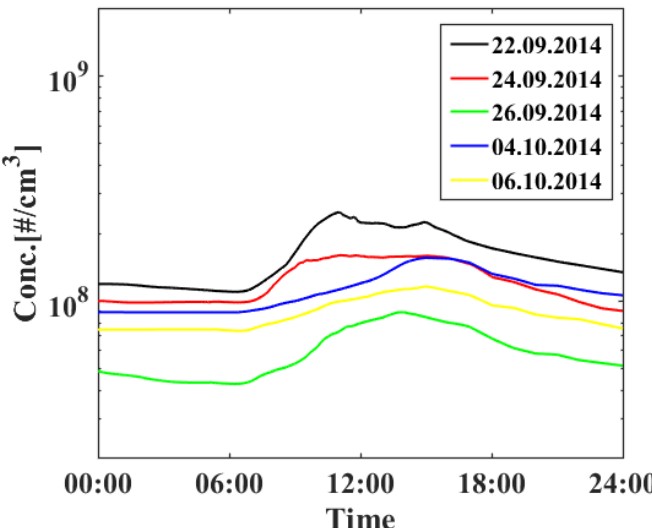

**Figure 3.** Simulated diurnal cycles of HOM formed from aromatics oxidation at SORPES on each chosen day.





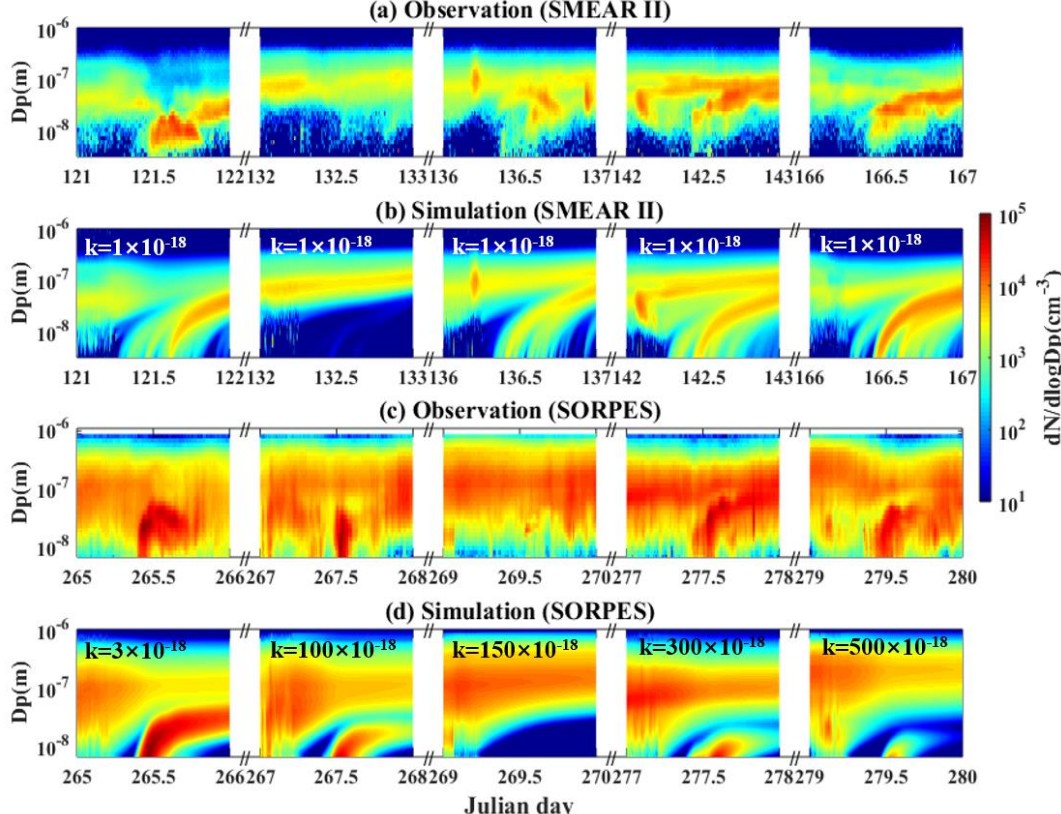

**Figure 4.** (a, c) Measured and (b, d) simulated particle number size distribution at SMEAR II and SORPES, respectively. Note: the kinetic coefficient on each day is texted in Figs. 2b & 2d.





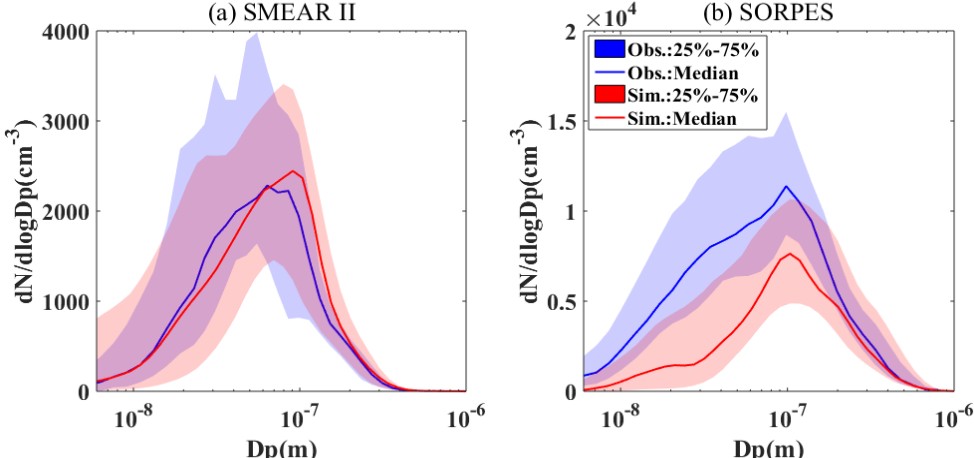

**Figure 5.** The observed and simulated aerosol number size distributions (a) at SMEAR II and (b) at SORPES. Note: Observed and simulated average (line) and ±1 standard deviation (shaded area) are in blue and red, respectively.





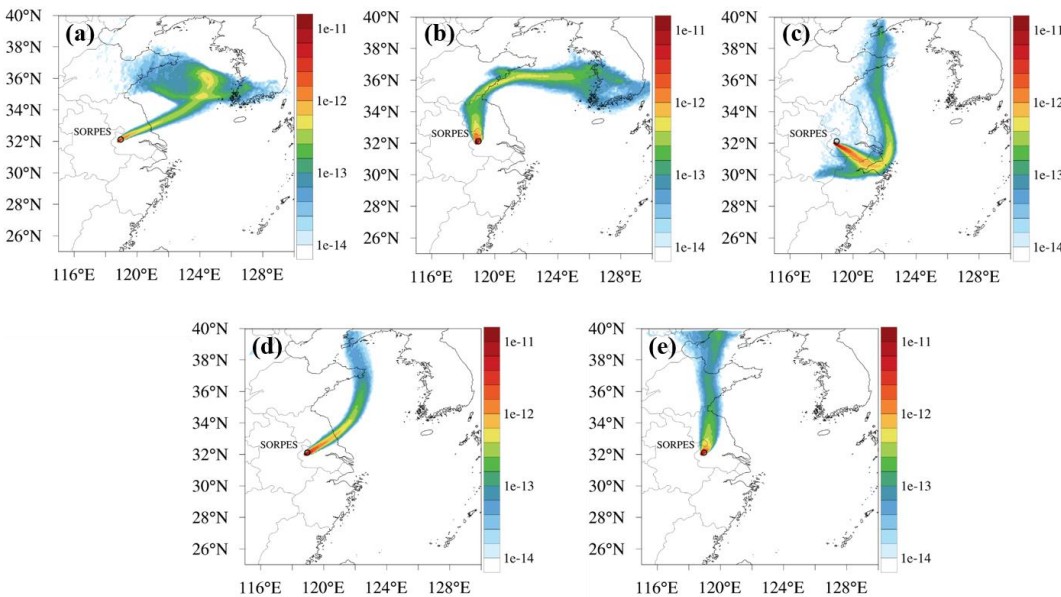

**Figure 6.** The averaged retroplume (footprint residence time) from 9:00 L.T. to 15:00 L.T. on (a) 22 September, (b) 24 September, (c) 26 September, (d) 4 October and (e) 6 October, 2014.





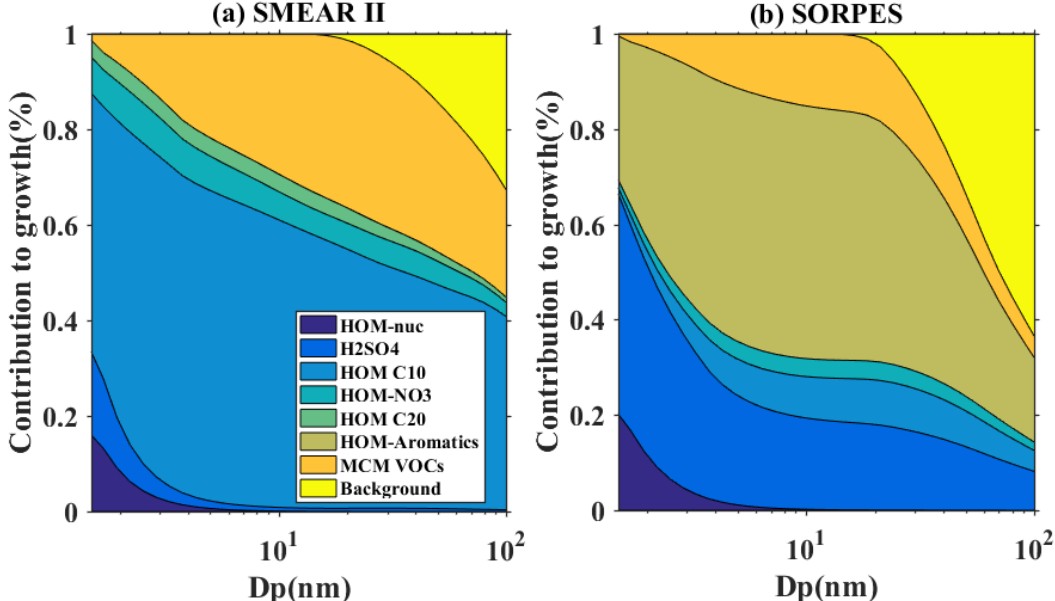

**Figure 7.** The relative contributions of precursor vapors to the growth of sub-100 nm particles at (a) SMEAR II and (b) SORPES.




**Table 1.** The statistics of observed formation rates of 6 nm particles ($J_6$), growth rates of 6-30 nm particles (GR) and condensation sinks (CS), $O_3$, $SO_2$, $NO_x$ concentrations from 9:00 LT to 15:00 LT on NPF days at SMEAR II and SORPES. Note: The statistical samples are the whole year database of 2013 at SMEAR II and the whole year database of 2014 at SORPES.

| | SMEAR II | | | | SORPES | | | |
|---|---|---|---|---|---|---|---|---|
| | Average | Median | 25th | 75th | Average | Median | 25th | 75th |
| $J_6$ ($cm^{-3}s^{-1}$) | 0.3 | 0.1 | 0.06 | 0.3 | 2.3 | 1.6 | 1 | 3.5 |
| GR (nm/h) | 4.5 | 2.8 | 2.0 | 5.6 | 8.7 | 8.0 | 6.5 | 10.4 |
| CS ($10^{-2}s^{-1}$) | 0.18 | 0.14 | 0.08 | 0.24 | 3.0 | 2.7 | 2.1 | 3.6 |
| $O_3$ (ppbv) | 36.1 | 36.6 | 29.6 | 41.8 | 44.7 | 43.3 | 28.0 | 59.1 |
| $SO_2$ (ppbv) | 0.2 | 0.1 | 0.03 | 0.3 | 9.4 | 8.0 | 4.4 | 12.7 |
| $NO_x$ (ppbv) | 0.5 | 0.2 | 0.06 | 0.6 | 17.7 | 13.4 | 7.9 | 23.0 |





**Table 2.** The NPF classification and environmental conditions on each chosen case day at SMEAR II and SORPES. Note: Condensation sink, meteorological conditions and the concentrations of trace gases are from 9:00 L.T. to 15:00 L.T.

| Case | NPF Classification | CS ($10^{-2}$ s$^{-1}$) | Temp (°C) | Rad (W/m$^2$) | RH (%) | O$_3$ (ppbv) | SO$_2$ (ppbv) | NO$_x$ (ppbv) | Mono (ppbv) | Benz. (ppbv) |
|------|------|------|------|------|------|------|------|------|------|------|
| **SMEAR II** | | | | | | | | | | |
| 05/01/2013 | NPF | 0.06 | 7.1 | 605.1 | 41.1 | 36.0 | 0.03 | 0.05 | 0.05 | 0.08 |
| 05/12/2013 | Non-NPF | 0.3 | 13.8 | 553.2 | 43.0 | 40.4 | 0.07 | 0.08 | 0.1 | 0.06 |
| 05/16/2013 | NPF | 0.3 | 17.6 | 682.9 | 27.9 | 53.2 | 0.05 | 0.1 | 0.2 | 0.05 |
| 05/22/2013 | NPF | 0.3 | 16.3 | 471.7 | 40.7 | 35.7 | 0.2 | 0.1 | 0.2 | 0.06 |
| 06/15/2013 | NPF | 0.1 | 14.8 | 486.6 | 59.0 | 32.3 | 0.04 | 0.07 | 0.1 | 0.04 |
| **SORPES** | | | | | | | | | | |
| 09/22/2014 | NPF | 2.1 | 24.6 | 497.0 | 60.2 | 45.2 | 2.4 | 7.7 | 0.04 | 0.7 |
| 09/24/2014 | NPF | 2.8 | 25.5 | 550.5 | 64.3 | 44.6 | 2.5 | 5.8 | 0.05 | 0.4 |
| 09/26/2014 | Non-NPF | 5.5 | 24.5 | 298.4 | 72.5 | 46.2 | 5.5 | 8.8 | 0.1 | 0.7 |
| 10/04/2014 | NPF | 2.5 | 22.2 | 567.6 | 53.7 | 36.2 | 8.3 | 22.2 | 0.04 | 0.6 |
| 10/06/2014 | NPF | 2.2 | 20.4 | 561.4 | 48.3 | 41.6 | 4.1 | 6.9 | 0.02 | 0.3 |





**Table 3.** The observed and simulated formation rates of 6 nm particles (J6) and growth rates of 6-30 nm particles (GR) on chosen NPF days at each site.

|  | $J_6$ obs. (cm$^{-3}$s$^{-1}$) | $J_6$ sim. (cm$^{-3}$s$^{-2}$) | GR obs. (nm/h) | GR sim. (nm/h) |
|---|---|---|---|---|
| **SMEAR II** |  |  |  |  |
| 05/01/2013 | 0.6 | 0.3 | 3.8 | 3.7 |
| 05/16/2013 | 0.06 | 0.07 | 3.3 | 3.6 |
| 05/22/2013 | 0.05 | 0.3 | 4.0 | 4.5 |
| 06/15/2013 | 0.08 | 0.6 | 5.2 | 4.8 |
| **SORPES** |  |  |  |  |
| 09/22/2014 | 4.9 | 5.6 | 9.9 | 7.8 |
| 09/24/2014 | 6.9 | 2.2 | 16.2 | 3.3 |
| 10/04/2014 | 3.8 | 1.8 | 14.9 | 2.8 |
| 10/06/2014 | 2.9 | 0.4 | 12.9 | 2.8 |