# Peer review of "Modelling studies of HOM and its contributions to new particle formation and growth: comparison of boreal forest in Finland and polluted environment in China"

_Atmospheric Chemistry and Physics, 2018_

## Referee Comment (RC1) · Anonymous Referee #1 · 7 Jun 2018

This is a very interesting Manuscript since it measures HOM within urban China and to investigate new particle formation mechanism at this polluted location. An important aspect is NPF is observed even at a polluted site with high condensation sink of vapors.

The manuscript is important for our understanding of NPF in polluted locations. Following are some questions which need to be addressed: 1. On page 8, the authors mention heterogeneous uptake of amines. Is it included in their model?

2. Why is SOA underpredicted at polluted locations? Is it because their model only

treats monoterpene SOA?

3. Can the authors comment on what other SOA types are likely to be important at SORPES? For example, anthropogenic SOA, isoprene SOA, IEPOX SOA etc.? What is isoprene concentration at SORPES (measured or predicted by WRF-Chem)?

4. Given the much higher 'k' value needed at SORPES compared to SMEAR II it is quite plausible that other NPF mechanisms are at play. Do the authors expect these NPF mechanisms to be dominated by anthropogenic processes?

5. Why does high Nox suppress new particle formation? Is it due to suppression of HOM dimers?

6. Can HOM organonitrates contribute to NPF/growth?

7. In addition to the formation of gas-phase HOM from aromatics oxidation, could aromatics oxidation contribute to HOM through heterogeneous uptake processes?

8. Can the authors comment on what measurements could be used to understand the relative roles of gas-phase and heterogeneous HOM processes on NPF/growth in polluted urban environments?

9. Seems a detailed modeling study that treats various SOA precursors and processes e.g. WRF-Chem at SORPES may be valuable for providing further insights into expected processes. Could the authors comment on how such a study could be used to augment their measurements and box model?

---

## Referee Comment (RC2) · Anonymous Referee #2 · 21 Jun 2018

This paper compares the new particle formation and growth processes in two contrasting atmospheric environments, a clean boreal forest and a polluted suburb with dominance of biogenic and anthropogenic emissions respectively. The results show that the MALTE-BOX model tended to significantly underestimate the particle growth at the polluted SORPES site. Sulfuric acid and HOMs produced from the oxidation of aromatics play a significant role in the particle growth at the polluted site, compared to the boreal forest site where the oxidation of monoterpenes is dominant. Overall, this study presents very interesting results regarding the new particle formation and growth, and

demonstrates the complexity and diversity of particle formation and growth processes at different atmospheric conditions. The manuscript is concisely organized and well written. Thus I suggest that this manuscript can be considered for publication after the following comments are addressed.

Specific comments:

Abstract: although the authors present a lot of interesting results in this paper, the abstract is too simple to cover all of the important results/findings of this study. The abstract should be informative enough and self-contained, and the readers can get all of the key points from only reading the abstract. The current abstract needs a significant revision to summarize the major results and conclusions of this study.

Section 2.1: some basic information is missing from the description of the measurement techniques. For example, what trace gas species and aerosol properties were specifically measured at both sites? What instruments were used for the aerosol size distribution measurements and what are the size ranges for them? What time periods of data were used in the analyses? It would be also helpful if the authors could provide a table to summarize the detailed observations including the species, measurement techniques, periods, etc. This will help the readers better navigate and understand the presented results.

Section 2.2: although the MALTE model has been described separately in some previous studies, a detailed description of this model is still needed for the present paper. The authors are suggested to provide further details about the model configuration, including chemistry mechanism, aerosol dynamic processes, mechanisms for HOM formation from monoterpene and aromatics oxidation, coagulation process, dry deposition and dilution processes, in the supplementary materials. Such a detailed description will help the readers better understand and reproduce the present study.

Typographical corrections and minor comments:

P1, L24: This study highlights...

P2, L9: define the "HOM".

P2, L13-14: a strong source of HOM and these...

P2, L28: one of the "flagship" stations...

P3, L1: the SORPES is defined as an urban site here, which is inconsistent with the sub-urban site as defined in the abstract and site description.

P5, L7: occurs even more frequently...

P5, L9: at SMEAR II is...

P5, L13-22: this paragraph compares the difference in the environmental conditions observed at the two study sites. It would be also helpful if the difference in the BVOC concentrations is mentioned here. The reviewer presumes that the BVOC levels at SMEAR II should be higher than at SORPES, although its concentrations of anthropogenic species are much lower.

P6, L31 – P7, L1: this statement seems to be not true as a simplified mechanism for the HOM formation from oxidation of aromatics was still included in the model.

P7, L24-27: the same to the above comment. The HOM formation from the oxidation of aromatics has been parameterized in the model.

Figure 1: provide the sources of the emission inventory data.

Figure 4 captions: it should be Figs. 4b & 4d?

Table 1: it would be helpful if the observed data for the other relevant species, such as VOCs, HOMs and meteorological parameters, were provided here, if the data are available.

---

## Author Comment (AC1) · 27 Jul 2018

**Response to the Referee #1's comments**

The authors deeply thank the reviewers' valuable comments, which will surely help improve the manuscript. The comments and responses are listed below:

1. On the page 8, the authors mention heterogeneous uptake of amines. Is it included in their model?
*Response: Thanks for the comment. In the current version of MALTE-BOX model, the heterogeneous uptake of amines hasn't been included. We mention this as only one of the possible reasons to explain the underestimation of growth rate here.*

2. Why is SOA underpredicted at polluted locations? Is it because their model only treats monoterpene SOA?
*Response: Besides the monoterpene formed SOA, the MALTE-BOX model also considers the isoprene and anthropogenic SOA, etc. However, the mechanisms of SOA formation especially for the anthropogenic SOA are still unclear and the current MALTE-BOX only considers aromatics SOA formation. Moreover, other anthropogenic gas vapors which are not considered in the modelling studies, may also contribute to the SOA. We will add some explanations to address this comment in section 3.3 in the revised manuscript.*

3. Can the authors comment on what other SOA types are likely to be important at SORPES? For example, anthropogenic SOA, isoprene SOA, IEPOX SOA etc.? What is isoprene concentration at SORPES (measured or predicted by WRF-Chem)?
*Response: The isoprene concentration was measured by GC-MS at SORPES in this study. Given such high anthropogenic VOCs at SORPES, anthropogenic SOA is one of the most important SOA at SORPES (Hu et al., 2017). In South China, the biogenic VOCs emission is quite high (Fig1 in manuscript) and the interactions between biogenic and anthropogenic emissions might play an important role in biogenic SOA formation (Zhang et al., 2017;Carlton et al., 2009). Therefore, isoprene SOA, including the isoprene epoxydiols (IEPOX) SOA can also be important at SORPES especially in summer when the air masses come from South China. We will add some descriptions to address this comment in the revised manuscript.*

4. Given the much higher 'k' value needed at SORPES compared to SMEAR II it is quite plausible that other NPF mechanisms are at play. Do the authors expect these NPF mechanisms to be dominated by anthropogenic processes?
*Response: Thanks for the comment. Yes, the k-values were much higher at SORPES than at SMEAR II, which means the nucleation mechanisms at two sites are different and only the biogenic processes are not enough for NPF at SORPES. The SMEAR II station is a typical boreal forest site and the SORPES station is a sub-urban site in the polluted eastern China. Therefore, it is highly possible that the anthropogenic processes that dominate the NPF in polluted region of China. We will add some descriptions to address this question in the revised manuscript.*

5. Why does high NOx suppress new particle formation? Is it due to suppression of HOM dimers?
*Response: Wildt et al. (2014) conducted the chamber experiments and found that NOx suppresses the NPF by reacting with the peroxy radicals(Wildt et al., 2014). NOx was found to suppress the*

*HOM dimers also by reacting with the peroxy radicals(Ehn et al., 2014). Given the HOM dimers have the lower volatile than HOM organonitrate, high NOx suppressing new particle formation might be due to the suppression of HOM dimers.*

6. Can HOM organonitrates contribute to NPF/growth?

***Response:*** *Yes, as some of HOM organonitrates are high oxidized and have low volatile, it is highly possible that they can contribute to NPF/growth.*

7. In addition to the formation of gas-phase HOM from aromatics oxidation, could aromatics oxidation contribute to HOM through heterogeneous uptake processes?

***Response:*** *It is possible that there are other mechanisms of HOM formation from aromatics oxidation. However, to the best of our knowledge, so far there are no works reported that the aromatics oxidation contributing to HOM through heterogeneous uptake processes yet.*

8. Can the authors comment on what measurements could be used to understand the relative roles of gas-phase and heterogeneous HOM processes on NPF/growth in polluted urban environments?

***Response:*** *Thanks. Long-term measurements of HOM by CI-APi-TOF, combined with relevant measurements covering the physicochemical properties of gaseous precursors, oxidants, clusters and aerosol particles can help understand the HOM formation and its roles on NPF in polluted urban environments. We will add these messages as an outlook in the discussion/conclusion part in the revision.*

9. Seems a detailed modeling study that treats various SOA precursors and processes e.g. WRF-Chem at SORPES may be valuable for providing further insights into expected processes. Could the authors comment on how such a study could be used to augment their measurements and box model?

***Response:*** *Thanks for the comments. Combining different models (e.g. WRF-Chem and MALTE-BOX) do provide further insights into expected processes, especially when the measurement didn't capture all relevant species (Huang et al., 2016). A comparison of the box-modeling results with measurements will provide us many important information to augment our measurements and the chemical mechanism in our box model. For example, we are going to add some measurements of some key species listed in the above comment (No.8) to improve our future understanding of NPF in this region. Also, after realizing that some missing anthropogenic processes are important for NPF in the region with strong anthropogenic impact, we are going to further develop the box model based on more quantitative chamber study in the future. We will add these messages as an outlook in the discussion/conclusion part in the revision.*

***References:***

*Carlton, A. G., Wiedinmyer, C., and Kroll, J. H.: A review of Secondary Organic Aerosol (SOA) formation from isoprene, Atmos. Chem. Phys., 9, 4987-5005, 10.5194/acp-9-4987-2009, 2009.*

*Ehn, M., Thornton, J. A., Kleist, E., Sipila, M., Junninen, H., Pullinen, I., Springer, M., Rubach, F., Tillmann, R., Lee, B., Lopez-Hilfiker, F., Andres, S., Acir, I. H., Rissanen, M., Jokinen, T., Schobesberger, S., Kangasluoma, J., Kontkanen, J., Nieminen, T., Kurten, T., Nielsen, L. B.,*

Jorgensen, S., Kjaergaard, H. G., Canagaratna, M., Dal Maso, M., Berndt, T., Petaja, T., Wahner, A., Kerminen, V. M., Kulmala, M., Worsnop, D. R., Wildt, J., and Mentel, T. F.: A large source of low-volatility secondary organic aerosol, Nature, 506, 476-+, 10.1038/nature13032, 2014.

Hu, J. L., Wang, P., Ying, Q., Zhang, H. L., Chen, J. J., Ge, X. L., Li, X. H., Jiang, J. K., Wang, S. X., Zhang, J., Zhao, Y., and Zhang, Y. Y.: Modeling biogenic and anthropogenic secondary organic aerosol in China, Atmos. Chem. Phys., 17, 77-92, 10.5194/acp-17-77-2017, 2017.

Huang , X , Zhou , L , Ding , A , Qi , X , Nie , W , Wang , M , Chi , X , Petäjä , T , Kerminen , V-M , Roldin , P , Rusanen , A , Kulmala , M & Boy , M 2016 , ' Comprehensive modelling study on observed new particle formation at the SORPES station in Nanjing, China ' Atmospheric Chemistry and Physics , vol 16 , no. 4 , pp. 2477-2492 . DOI: 10.5194/acp-16-2477-2016

Wildt, J., Mentel, T. F., Kiendler-Scharr, A., Hoffmann, T., Andres, S., Ehn, M., Kleist, E., Musgen, P., Rohrer, F., Rudich, Y., Springer, M., Tillmann, R., and Wahner, A.: Suppression of new particle formation from monoterpene oxidation by NOx, Atmos. Chem. Phys., 14, 2789-2804, 10.5194/acp-14-2789-2014, 2014.

Zhang, Y. J., Tang, L. L., Sun, Y. L., Favez, O., Canonaco, F., Albinet, A., Couvidat, F., Liu, D. T., Jayne, J. T., Wang, Z., Croteau, P. L., Canagaratna, M. R., Zhou, H. C., Prevot, A. S. H., and Worsnop, D. R.: Limited formation of isoprene epoxydiols-derived secondary organic aerosol under NOx-rich environments in Eastern China, Geophys. Res. Lett., 44, 2035-2043, 10.1002/2016gl072368, 2017.

**Response to the Referee #2's comments**

The authors deeply thank the reviewers' comments, which will surely improve the manuscript. The comments and responses are listed below:

Specific comments:

Abstract: although the authors present a lot of interesting results in this paper, the abstract is too simple to cover all of the important results/findings of this study. The abstract should be informative enough and self-contained, and the readers can get all of the key points from only reading the abstract. The current abstract needs a significant revision to summarize the major results and conclusions of this study.

*Response: Thanks for the comment. We will rewrite the abstract to summarize the major results and conclusions.*

Section 2.1: some basic information is missing from the description of the measurement techniques. For example, what trace gas species and aerosol properties were specifically measured at both sites? What instruments were used for the aerosol size distribution measurements and what are the size ranges for them? What time periods of data were used in the analyses? It would be also helpful if the authors could provide a table to summarize the detailed observations including the species, measurement techniques, periods, etc. This will help the readers better navigate and understand the presented results.

*Response: Thanks for the valuable suggestion. We will add those descriptions of the measurement techniques and provide a table to list all the information of observations in the manuscript.*

Section 2.2: although the MALTE model has been described separately in some previous studies, a detailed description of this model is still needed for the present paper. The authors are suggested to provide further details about the model configuration, including chemistry mechanism, aerosol dynamic processes, mechanisms for HOM formation from monoterpene and aromatics oxidation, coagulation process, dry deposition and dilution processes, in the supplementary materials. Such a detailed description will help the readers better understand and reproduce the present study.

*Response: Thanks for the suggestion. We will describe the MALTE model in details in the supplementary materials.*

Typographical corrections and minor comments:

P1, L24: This study highlights…

*Response: We will correct it in the revised version.*

P2, L9: define the "HOM".

*Response: We will define the "HOM" in revised version.*

P2, L13-14: a strong source of HOM and these…

*Response: We will correct it in the revised version.*

P2, L28: one of the "flagship" stations…

*Response: We will correct it in the revised version.*

P3, L1: the SORPES is defined as an urban site here, which is inconsistent with the sub-urban site as defined in the abstract and site description.
*Response: We will correct it into 'a sub-urban site' here.*

P5, L7: occurs even more frequently…
*Response: We will correct it in the revised version.*

P5, L9: at SMEAR II is…
*Response: We will correct it in the revised version.*

P5, L13-22: this paragraph compares the difference in the environmental conditions observed at the two study sites. It would be also helpful if the difference in the BVOC concentrations is mentioned here. The reviewer presumes that the BVOC levels at SMEAR II should be higher than at SORPES, although its concentrations of anthropogenic species are much lower.
*Response: Thanks for the suggestion. We will add the discussions of the difference in the BVOC concentrations at the two station in revised version.*

P6, L31 – P7, L1: this statement seems to be not true as a simplified mechanism for the HOM formation from oxidation of aromatics was still included in the model.
*Response: We will change this statement in the revised version.*

P7, L24-27: the same to the above comment. The HOM formation from the oxidation of aromatics has been parameterized in the model.
*Response: We will correct this statement in the revised version.*

Figure 1: provide the sources of the emission inventory data.
*Response: We will provide the sources of the emission inventory data in revised version.*

Figure 4 captions: it should be Figs. 4b & 4d?
*Response: We will correct it in the revised version.*

Table 1: it would be helpful if the observed data for the other relevant species, such as VOCs, HOMs and meteorological parameters, were provided here, if the data are available.
*Response: Thanks for the suggestion. We will provide information of all used available measurement parameters in Table 1 in revised version.*

---

## Author Response (AR1)

**Response to the Referee #1's comments**

The authors deeply thank the reviewers' valuable comments, which surely help improve the manuscript. The comments and responses are listed below:

1.  On the page 8, the authors mention heterogeneous uptake of amines. Is it included in their model?
*Response: Thanks for the comment. In the current version of MALTE-BOX model, the heterogeneous uptake of amines hasn't been included. We mentioned this as only one of the possible reasons to explain the underestimation of growth rate in P8, L25-26 of the revised version.*

2.  Why is SOA underpredicted at polluted locations? Is it because their model only treats monoterpene SOA?
*Response: Besides the monoterpene formed SOA, the MALTE-BOX model also considers the isoprene and anthropogenic SOA, etc. However, the mechanisms of SOA formation especially for the anthropogenic SOA are still unclear and other anthropogenic gas vapors which are not considered in the modelling studies, may also contribute to the SOA. We added some explanations to address this comment in P9, L32-P10, L3 in the revised manuscript.*

3.  Can the authors comment on what other SOA types are likely to be important at SORPES? For example, anthropogenic SOA, isoprene SOA, IEPOX SOA etc.? What is isoprene concentration at SORPES (measured or predicted by WRF-Chem)?
*Response: The isoprene concentration was measured by GC-MS at SORPES in this study. Given such high anthropogenic VOCs at SORPES, anthropogenic SOA is one of the most important SOA at SORPES (Hu et al., 2017). In South China, the biogenic VOCs emission is quite high (Fig1 in manuscript) and the interactions between biogenic and anthropogenic emissions might play an important role in biogenic SOA formation (Zhang et al., 2017;Carlton et al., 2009). Therefore, isoprene SOA, including the isoprene epoxydiols (IEPOX) SOA can also be important at SORPES especially in summer when the air masses come from South China. We added some descriptions to address this comment in P6, L6-9 the revised manuscript.*

4.  Given the much higher 'k' value needed at SORPES compared to SMEAR II it is quite plausible that other NPF mechanisms are at play. Do the authors expect these NPF mechanisms to be dominated by anthropogenic processes?
*Response: Thanks for the comment. Yes, the k-values were much higher at SORPES than at SMEAR II, which means the nucleation mechanisms at two sites are different and only the biogenic processes are not enough for NPF at SORPES. The SMEAR II station is a typical boreal forest site and the SORPES station is a sub-urban site in the polluted eastern China. Therefore, it is highly possible that the anthropogenic processes that dominate the NPF in polluted region of China. We added some descriptions to address this question in P8, L12-14 of the revised manuscript.*

5.  Why does high NOx suppress new particle formation? Is it due to suppression of HOM dimers?
*Response: Wildt et al. (2014) conducted the chamber experiments and found that NOx suppresses the NPF by reacting with the peroxy radicals(Wildt et al., 2014). NOx was found to suppress the HOM dimers also by reacting with the peroxy radicals(Ehn et al., 2014). Given the HOM dimers*

*have the lower volatile than HOM organonitrate, high NOx suppressing new particle formation might be due to the suppression of HOM dimers.*

6. Can HOM organonitrates contribute to NPF/growth?

***Response:*** *Yes, as some of HOM organonitrates are high oxidized and have low volatile, it is highly possible that they can contribute to NPF/growth.*

7. In addition to the formation of gas-phase HOM from aromatics oxidation, could aromatics oxidation contribute to HOM through heterogeneous uptake processes?

***Response:*** *It is possible that there are other mechanisms of HOM formation from aromatics oxidation. However, to the best of our knowledge, so far there are no works reported that the aromatics oxidation contributing to HOM through heterogeneous uptake processes yet.*

8. Can the authors comment on what measurements could be used to understand the relative roles of gas-phase and heterogeneous HOM processes on NPF/growth in polluted urban environments?

***Response:*** *Thanks. Long-term measurements of HOM by CI-APi-TOF, combined with relevant measurements covering the physicochemical properties of gaseous precursors, oxidants, clusters and aerosol particles can help understand the HOM formation and its roles on NPF in polluted urban environments. We added these messages as an outlook in the conclusion part in P10, L26-P11, L4 of the revised version.*

9. Seems a detailed modeling study that treats various SOA precursors and processes e.g. WRF-Chem at SORPES may be valuable for providing further insights into expected processes. Could the authors comment on how such a study could be used to augment their measurements and box model?

***Response:*** *Thanks for the comments. Combining different models (e.g. WRF-Chem and MALTE-BOX) do provide further insights into expected processes, especially when the measurement didn't capture all relevant species (Huang et al., 2016). A comparison of the box-modeling results with measurements will provide us many important information to augment our measurements and the chemical mechanism in our box model. For example, we are going to add some measurements of some key species listed in the above comment (No.8) to improve our future understanding of NPF in this region. Also, after realizing that some missing anthropogenic processes are important for NPF in the region with strong anthropogenic impact, we are going to further develop the box model based on more quantitative chamber study in the future. We added these messages as an outlook in the conclusion part in P10, L26-P11, L4 of the revised version.*

**Response to the Referee #2's comments**

The authors deeply thank the reviewers' comments, which surely improve the manuscript. The comments and responses are listed below:

Specific comments:

Abstract: although the authors present a lot of interesting results in this paper, the abstract is too simple to cover all of the important results/findings of this study. The abstract should be informative enough and self-contained, and the readers can get all of the key points from only reading the abstract. The current abstract needs a significant revision to summarize the major results and conclusions of this study.

*Response: Thanks for the comment. We rewrote the abstract to summarize the major results and conclusions in revised manuscript.*

Section 2.1: some basic information is missing from the description of the measurement techniques. For example, what trace gas species and aerosol properties were specifically measured at both sites? What instruments were used for the aerosol size distribution measurements and what are the size ranges for them? What time periods of data were used in the analyses? It would be also helpful if the authors could provide a table to summarize the detailed observations including the species, measurement techniques, periods, etc. This will help the readers better navigate and understand the presented results.

*Response: Thanks for the valuable suggestion. We added those descriptions of the measurement techniques in P3, L18-24 and provided a table (Table S1) to list all the information of observations in supplementary materials.*

Section 2.2: although the MALTE model has been described separately in some previous studies, a detailed description of this model is still needed for the present paper. The authors are suggested to provide further details about the model configuration, including chemistry mechanism, aerosol dynamic processes, mechanisms for HOM formation from monoterpene and aromatics oxidation, coagulation process, dry deposition and dilution processes, in the supplementary materials. Such a detailed description will help the readers better understand and reproduce the present study.

*Response: Thanks for the suggestion. We described the MALTE model in details in the supplementary materials (Appendix B).*

Typographical corrections and minor comments:

P1, L24: This study highlights…

*Response: We corrected it in the revised version.*

P2, L9: define the "HOM".

*Response: We defined the "HOM" in revised version.*

P2, L13-14: a strong source of HOM and these…

*Response: We corrected it in the revised version.*

P2, L28: one of the "flagship" stations…
*Response: We corrected it in the revised version.*

P3, L1: the SORPES is defined as an urban site here, which is inconsistent with the sub-urban site as defined in the abstract and site description.
*Response: We corrected it into 'a sub-urban site' here.*

P5, L7: occurs even more frequently…
*Response: We corrected it in the revised version.*

P5, L9: at SMEAR II is…
*Response: We corrected it in the revised version.*

P5, L13-22: this paragraph compares the difference in the environmental conditions observed at the two study sites. It would be also helpful if the difference in the BVOC concentrations is mentioned here. The reviewer presumes that the BVOC levels at SMEAR II should be higher than at SORPES, although its concentrations of anthropogenic species are much lower.
*Response: Thanks for the suggestion. We added the discussions of the difference in the BVOC concentrations at the two stations in P5, L30-31 revised version.*

P6, L31 – P7, L1: this statement seems to be not true as a simplified mechanism for the HOM formation from oxidation of aromatics was still included in the model.
*Response: We changed this statement in the revised version.*

P7, L24-27: the same to the above comment. The HOM formation from the oxidation of aromatics has been parameterized in the model.
*Response: We corrected this statement in the revised version.*

Figure 1: provide the sources of the emission inventory data.
*Response: We provided the sources of the emission inventory data in revised version.*

Figure 4 captions: it should be Figs. 4b & 4d?
*Response: We corrected it in the revised version.*

Table 1: it would be helpful if the observed data for the other relevant species, such as VOCs, HOMs and meteorological parameters, were provided here, if the data are available.
*Response: Thanks for the suggestion. We provided information of all used available measurement parameters in Table 1 in revised version.*

[revised manuscript text omitted]